# Parity-Time Symmetric Holographic Principle

**DOI:** 10.3390/e25111523

**Published:** 2023-11-07

**Authors:** Xingrui Song, Kater Murch

**Affiliations:** Department of Physics, Washington University, St. Louis, MO 63130, USA; songx@wustl.edu

**Keywords:** parity-time symmetry, holographic principal, quantum simulation

## Abstract

Originating from the Hamiltonian of a single qubit system, the phenomenon of the avoided level crossing is ubiquitous in multiple branches of physics, including the Landau–Zener transition in atomic, molecular, and optical physics, the band structure of condensed matter physics and the dispersion relation of relativistic quantum physics. We revisit this fundamental phenomenon in the simple example of a spinless relativistic quantum particle traveling in (1+1)-dimensional space-time and establish its relation to a spin-1/2 system evolving under a PT-symmetric Hamiltonian. This relation allows us to simulate 1-dimensional eigenvalue problems with a single qubit. Generalizing this relation to the eigenenergy problem of a bulk system with *N* spatial dimensions reveals that its eigenvalue problem can be mapped onto the time evolution of the edge state with (N−1) spatial dimensions governed by a non-Hermitian Hamiltonian. In other words, the bulk eigenenergy state is encoded in the edge state as a hologram, which can be decoded by the propagation of the edge state in the temporal dimension. We argue that the evolution will be PT-symmetric as long as the bulk system admits parity symmetry. Our work finds the application of PT-symmetric and non-Hermitian physics in quantum simulation and provides insights into the fundamental symmetries.

## 1. Introduction

In the 1980s, Richard Feynman envisioned the advantage of using quantum mechanical systems to simulate quantum physics [1]. As most formulations of quantum mechanics consider the systems to be governed by Hermitian Hamiltonians, the community established the theory of quantum computation based on combinations of unitary gate operations [2,3,4,5]. However, after decades of effort toward implementing quantum computers, we realize that even the most highly controlled quantum systems are open quantum systems [6,7,8]. These open quantum systems are non-unitary, suffering from the residual coupling with the environment, causing dissipation and decoherence. In contrast to the stereotype that such non-unitary effects of evolution are always harmful, dissipation is now considered an important resource for quantum technologies, with extensive applications in quantum control, sensing, and simulation [9,10,11,12,13,14,15,16,17]. In this letter, we extend Feynman’s logic by taking advantage of open quantum systems as an efficient resource for quantum simulation. In particular, we show how the open quantum system evolution described by a non-Hermitian Hamiltonian [15,18,19,20,21,22,23,24,25,26,27,28,29,30] allows one to map the eigenvalue problem of a 1-dimensional Hermitian system onto the time evolution of a qubit. Extending this reasoning suggests that an *N*-dimensional Hermitian bulk system can be mapped onto the non-Hermitian time evolution of a (N−1)-dimensional edge system. In parallel with the research highlighting the bulk-boundary correspondence of non-Hermitian systems [31,32,33,34,35], our work finds the new application of these concepts to reduce the quantum resources required for encoding the spatial degrees of freedom in quantum simulation for Hermitian systems. We present two specific examples that illustrate the power of this concept. First, we propose an experimental scheme exploiting this relationship to perform a quantum simulation of a scattering problem with reduced quantum resources. Second, we examine the system of a Kitaev chain and show how the non-Hermitian evolution can model the physics of Majorana zero modes in one of the simplest topologically nontrivial models [36,37,38,39,40]. After demonstrating our method with two examples, we discuss the relation between Parity symmetry and Parity-Time (PT)-symmetry and generalize the result to show how PT-symmetric evolution of the edge state generates the solution of the eigenvalue problem of a bulk Hermitian system.

## 2. The Eigenvalue Problem and PT-Symmetric Evolution

We start by considering the simple case of a massless Fermion moving in one dimension, which has a linear dispersion relation as shown in Figure 1a. The right and left moving particles form a two-state system {|L〉,|R〉}, which can be mapped onto a qubit. In this gapless limit, the Dirac equation is uncoupled. Once the two components are coupled, the particle acquires a mass resulting in an avoided crossing (Figure 1b). In this case, the particle at rest is given by an equal superposition of the states |L〉 and |R〉. In general, the state of a traveling particle is given by some linear combination of |L〉 and |R〉. Particularly, the state $|\pm \rangle =\frac{1}{\sqrt{2}}\left(|L\rangle \pm |R\rangle \right)$ represents the particle at rest with eigenenergy equal to ±m. This picture implicitly incorporates the idea of separation of variables, where we downgrade the momentum from an operator to a parameter $-i\partial_{x}\to k$. Hence, we can write the corresponding Schrödinger equation for the particle with the time coordinate treated as the independent variable,
(1)$$i\partial_{t}\left(\begin{array}{c} \psi_{L}\\ \psi_{R} \end{array}\right)=\left(\begin{array}{cc} -k & m\\ m & k \end{array}\right)\left(\begin{array}{c} \psi_{L}\\ \psi_{R} \end{array}\right)=H\left(k\right)\left(\begin{array}{c} \psi_{L}\\ \psi_{R} \end{array}\right),$$
where we have set ℏ=c=1 for simplicity. This approach, however, is not the only option, and may not be the most convenient approach under some circumstances. Indeed, the Dirac equation does not prefer a specific time or spatial coordinate.

Instead, we consider replacing the energy, ω, with a parameter and treat the spatial coordinate as the independent variable,
(2)$$i\partial_{x}\left(\begin{array}{c} \psi_{L}\\ \psi_{R} \end{array}\right)=\left(\begin{array}{cc} \omega & -m\\ m & -\omega \end{array}\right)\left(\begin{array}{c} \psi_{L}\\ \psi_{R} \end{array}\right)=H_{\mathrm{eff}}\left(\omega \right)\left(\begin{array}{c} \psi_{L}\\ \psi_{R} \end{array}\right).$$
The matrix that appears here as an effective Hamiltonian, $H_{\mathrm{eff}}$, is clearly non-Hermitian, and while it is not obviously of a PT-symmetric form, it embodies the well-known concept of the PT-symmetry breaking transition [18,20,21,22,23,24,25,26,27,29,30], as is shown in Figure 2. Solving the characteristic polynomial for the eigenvalues *k* of this effective Hamiltonian,
(3)$$k=\pm \sqrt{\omega^{2}-m^{2}}$$
we can see that if ω is a real number, the solution for *k* can alternatively be real, or imaginary. This embodies the regions of, respectively, unbroken (Figure 2a) and broken PT-symmetry (Figure 2c), with an exceptional point (EP) occurring for |ω|=m (Figure 2b) [15,23,27,28,41,42]. Additionally, since $H_{\mathrm{eff}}$ is non-Hermitian, its eigenvectors are non-orthogonal.

This establishes the principle that we employ to harness quantum evolution in a resource-efficient manner for quantum simulation: we now let the laboratory time coordinate represent the spatial coordinate of Equation (Equation 2), such that the real-time evolution of a qubit captures the spatial solution of a chosen problem [43]. We refer to this principle as the *qubit hologram* because the qubit encodes the wavefront of the spatial solution. In general, we are interested in the scattering or bound states of a given potential V(x), which can now be obtained from the time evolution under,
(4)$$H\left(t\right)=\left(\begin{array}{cc} E-V(t) & -m\\ m & -E+V(t) \end{array}\right),$$
where we have replaced ω with *E* to emphasize it now represents the eigenenergy of the problem.

## 3. Example 1: Scattering Phase Shifts

The problem of solving the cross-section of an elastic neutron scattering off an atomic nucleus described by an optical potential can be reduced to a 1D problem through partial wave decomposition (Figure 3a) [44], which gives a differential equation for the radial wave function $u_{lj}\left(r\right)$ for the orbital angular momentum quantum number *l* and total angular momentum number *j*,
(5)$$u_{lj}^{\prime \prime }\left(r\right)+\left[k^{2}-\frac{2m}{\hbar^{2}}V_{lj}\left(r\right)-\frac{l(l+1)}{r^{2}}\right]u_{lj}\left(r\right)=0,$$
where $V_{lj}\left(r\right)$ is given by the global optical model CH89 [45]. We will show how this differential equation can be mapped onto the temporal dynamics of a qubit evolving in a time-dependent Hamiltonian. For this purpose, we find it convenient to work in the |±〉 basis. The time evolution of a qubit state |ψ〉 expressed in the basis |ψ〉=α|+〉+β|−〉, is given by a Hamiltonian *H*, which can be represented as a 2 × 2 matrix.

By introducing a transformation between the spatial and time coordinates r=vt, where *v* is a scaling parameter, we relate the qubit state |ψ〉 with the solution of the wavefunction, $u_{lj}\left(r\right)$ by writing,
(6)$$\begin{array}{cc} \alpha (t) & =\sqrt{2}u_{lj}\left(r\right), \end{array}$$
(7)$$\begin{array}{cc} \beta (t) & =i\partial_{r}u_{lj}\left(r\right)/\left(\sqrt{2}m\right), \end{array}$$
such that the expansion coefficients α(t),β(t) encode the wave function and its derivative.

To capture the solution to the differential Equation (Equation 5) as temporal dynamics, we need the expansion coefficients to obey the differential equations,
(8)$$\begin{array}{cc} i\partial_{t}\alpha \left(t\right) & =2vm\beta (t), \end{array}$$
(9)$$\begin{array}{cc} i\partial_{t}\beta \left(t\right) & =v(E-V_{\mathrm{tot}}\left(vt\right))\alpha \left(t\right), \end{array}$$
where
(10)$$V_{\mathrm{tot}}\left(r\right)=V_{lj}\left(r\right)+\frac{\hbar^{2}}{2m}\frac{l(l+1)}{r^{2}},$$
is the total potential, including the centrifugal term. The evolution described above can be implemented by a Hamiltonian written as
(11)$$\begin{array}{cc} \; & H_{\mathrm{eff}}\left(t\right)=v\left(\begin{array}{cc} 0 & 2m\\ E-V_{\mathrm{tot}}\left(vt\right) & 0 \end{array}\right), \end{array}$$
which is non-Hermitian. A non-Hermitian Hamiltonian such as Equation (Equation 11) can be obtained, for example, through the evolution of a dissipative qubit where post-selection is used to eliminate quantum jumps [18,19,46,47]. Another approach for synthesizing non-Hermitian dynamics is based on Hamiltonian dilation. However, this method requires additional computational resources.

Figure 3 displays a specific example where the evolution of a qubit is used to solve for the first partial wave channel (l=0,j=1/2) scattering of a neutron off of a ${}^{208}$Pb nucleus. Figure 3b displays the potential of the nucleus, where the imaginary part of the potential corresponds to the process of neutron capture. We calculate the time evolution of a qubit evolving under Equation (Equation 11) using QuTiP [48,49]. Figure 3c displays the resulting time evolution of the qubit, initialized in the state |−〉, from which we determine the partial wave solution $u_{lj}\left(r\right)$ (Figure 3d). By comparing the final phase to the solution for the free particle, we can determine the scattering phase shift and the scattering cross section (see Appendix A) [44]. By evaluating a series of these partial waves, we can sum up their scattering contributions and compare the result to the experimental data [45] as shown in Figure 3e. Here, we emphasize that the summation of partial waves can be executed on a classical computer. Consequently, the quantum resources needed for the scattering simulation are reduced to a single qubit in this example.

## 4. Example 2: Majorana Zero Mode

In addition to scattering states, which allow a continuum of eigenenergy values, our formalism also applies to bound states. In particular, topological bound states are a class of bound states with rich physics. We will show how our method is able to solve for such states, and how the PT-symmetry transition signals a topological phase transition. Here, we demonstrate a Majorana zero mode as an edge state of the Kitaev chain, which is a simplified model of a topological superconductor [36]. The realistic version has been experimentally demonstrated in semiconducting nanowires in proximity to superconductors [50,51,52,53].

The band structure of a condensed matter system describes the behavior of its excitations. Under perturbations, the band structure can be deformed. For a gapped system such as an insulator, a finite gap separates the bands. For small perturbations, this gap cannot be closed. As a result, for insulators, continuous perturbation induces continuous deformation of the band structure, yet these deformations are topologically equivalent. If a region of the normal band structure is connected to a region of inverted band structure, indicated by negative gap energy, the gap is forced to close, giving rise to a bound state. This gap closure is topologically protected, resulting in topologically protected bound states. In the case of a topological superconductor, the edge state is a Majorana zero mode.

The two topological phases that enclose the bound state correspond to regions of effective positive and negative mass, which we model as m(x)=−tanh(x). To calculate the bound states of this system, we again replace the spatial coordinate with the temporal coordinate such that the bound state appears in the time evolution of the effective Hamiltonian,
(12)$$H_{\mathrm{eff}}\left(t\right)=\left(\begin{array}{cc} E & -m(t)\\ m(t) & -E \end{array}\right),$$
where m(t) is the mass of the excitation that switches sign from +1 to −1 during the evolution (Figure 4a). *E* is the trial solution to the eigenvalue problem.

For the eigenvalue problem of bound states, the method works as a variational eigensolver. The key observation is the wave function of a bound state is an evanescent wave outside of the interface. This physical constraint guarantees its amplitude to vanish at sufficiently large distances. In our case, we focus on the regime with E≪m(t=0), where the Hamiltonian is dominated by the $\sigma_{y}$ component. We initialize the qubit in the $|y-\rangle =\frac{1}{\sqrt{2}}\left(|L\rangle -i|R\rangle \right)$ eigenstate of the non-Hermitian Hamiltonian when m=1, featuring an exponential gain behavior. Next, we let the system evolve under the non-Hermitian Hamiltonian to obtain the corresponding trial wavefunction (Figure 4b). We repeat the process with various trial eigenenergies and try to make the final state converge to an exponentially decaying state for the non-Hermitian Hamiltonian when m=−1. This approach is commonly known as the “shooting” method for finding solutions to differential equations [54]; here, the propagation is carried out by the time evolution of the qubit. Figure 4c displays the final amplitude (at x=5) as a function of *E* showing that the bound state occurs at E=0 as expected. In summary, we have represented the spatial wave function of a Majorana zero mode using just a single qubit, minimizing the quantum resources required for simulating a condensed matter system.

Now, we explain why PT-symmetric physics is important here. For a given *E* there are typically two classical turning points, which are the EPs of $H_{\mathrm{eff}}$. The EPs separate regions of broken and unbroken PT-symmetry corresponding, respectively, to exponential or oscillatory solutions. This is similar to what one encounters in the study of bound states in elementary quantum mechanics. Here, with no external potential, the choice of *E* specifies the EPs, and in particular when E→0 the two EPs merge. The merging of the two EPs corresponds to the topological phase transition that produces the Majorana bound state.

We have so far discussed two examples that show how the non-Hermitian time evolution of a qubit can be used to emulate the spatial solutions of scattering and bound state systems. This is the basis of the qubit hologram; in higher than one dimension, the quantum state in the bulk can be encoded in the time evolution of the edge qubit hologram. Additionally, the qubit hologram will inherit the P symmetry from the bulk system as PT-symmetry. We demonstrate the concept with a spin-1/2 particle moving in three dimensions governed by the Dirac Hamiltonian
(13)$$H\left(k\right)=\left(\begin{array}{cc} -\sigma \cdot k & m\\ m & \sigma \cdot k \end{array}\right),$$
which is a 4 × 4 matrix written in the block form. We choose to work in the momentum representation for the simplicity of the ensuing expressions. σ represents the Pauli matrices and k represents the momentum. The Dirac Hamiltonian conserves parity
(14)$$P^{-1}H\left(k\right)P=H\left(-k\right),\;P=-P^{-1}=i\left(\begin{array}{cc} \; & I\\ I \end{array}\right),$$
where we have included the prefactor *i* in P to make it comply with the parity of multi-particle wavefunctions [55]. P can be decomposed into three separate mirror reflection operators
(15)$$P=M_{x}M_{y}M_{z},\;M_{j}=M_{j}^{-1}=\left(\begin{array}{cc} \; & \sigma_{j}\\ \sigma_{j} \end{array}\right),$$
where j=x,y,z. The qubit hologram allows us to treat the *z* coordinate as the effective time axis, which yields the effective Hamiltonian,
(16)$$\begin{array}{cc} \; & H_{\mathrm{eff}}\left(\omega ,k_{x},k_{y}\right)=\\ \; & \left(\begin{array}{cc} \omega \sigma_{z}+i\left(k_{x}\sigma_{y}-k_{y}\sigma_{x}\right) & -m\sigma_{z}\\ m\sigma_{z} & -\omega \sigma_{z}+i\left(k_{x}\sigma_{y}-k_{y}\sigma_{x}\right) \end{array}\right). \end{array}$$
Since the initial system of the Dirac Hamiltonian is parity symmetric, the qubit hologram (Equation 16) now obeys PT-symmetry.
(17)$$P_{\mathrm{eff}}T_{\mathrm{eff}}H_{\mathrm{eff}}\left(\omega ,k_{x},k_{y}\right)P_{\mathrm{eff}}T_{\mathrm{eff}}=H_{\mathrm{eff}}\left(\omega ,-k_{x},-k_{y}\right),$$
where the time reversal and the parity operations are characterized by
(18)$$T_{\mathrm{eff}}=M_{z}=\left(\begin{array}{cc} \; & \sigma_{z}\\ \sigma_{z} \end{array}\right),P_{\mathrm{eff}}=M_{x}M_{y}=i\left(\begin{array}{cc} \sigma_{z} & \\ \; & \sigma_{z} \end{array}\right),$$
respectively. We note two subtleties regarding this representation: (i) Here, $T_{\mathrm{eff}}$ is a unitary operator (inherited from $M_{z}$) rather than an anti-unitary operator which involves complex conjugation and is typically associated with time reversal. (ii) The $P_{\mathrm{eff}}T_{\mathrm{eff}}$ operator transforms the original Hamiltonian with momentum $\left(k_{x},k_{y}\right)$ into a different Hamiltonian with spatially inverted momentum $\left(-k_{x},-k_{y}\right)$. For this reason, the PT-symmetry here appears to be different than what is typically encountered in the literature. The second subtlety can be addressed by considering the coordinate representation of the Hamiltonian or an extended momentum representation containing both $\left(k_{x},k_{y}\right)$ and $\left(-k_{x},-k_{y}\right)$. Furthermore, by redefining the complex numbers and the associated complex conjugation of the matrix elements, the anti-unitary nature of $T_{\mathrm{eff}}$ can be restored.

## 5. Conclusions


Many of the leading technologies for quantum processors consist of a fixed number of qubits that evolve in real-time with an evolution that can be characterized by a sequence of single and multi-qubit gates and ongoing effects of decoherence. In contrast, for applications of quantum simulation, we are typically interested in static properties with spatial structure, such as bound states, scattering cross-sections, and spatial correlations. Therefore, to effectively utilize a quantum simulator, it is desirable to understand how the time axis can be used to capture spatial structure. To achieve this, we have introduced a proper space-to-time transform—the qubit hologram—that is able to assign one of the spatial dimensions a new meaning of an effective time axis. This can have an impact on reducing resources for quantum simulation. For example, if we are interested in simulating the ground states of a M×M lattice of particles (e.g., qubits), then nominally, we would need $M^{2}$ qubits for the simulation. With this approach, we can reduce the resources to *M* qubits and time evolution, which is a huge reduction in qubit count simulation.

In nature, many physical systems can be described by Hamiltonians that respect P symmetry. Examples include field theories such as Quantum Chromodynamics and crystals with inversion symmetry, such as graphene. As we have seen, the eigenvalue problem for systems with P symmetry will map onto PT-symmetric dynamics of the system’s qubit hologram. Here, the real-space solutions corresponding to traveling waves, evanescent waves, and particles at rest map onto the familiar unbroken region, broken region, and exceptional points of PT-symmetry.

## Figures and Tables

**Figure 1 entropy-25-01523-f001:**
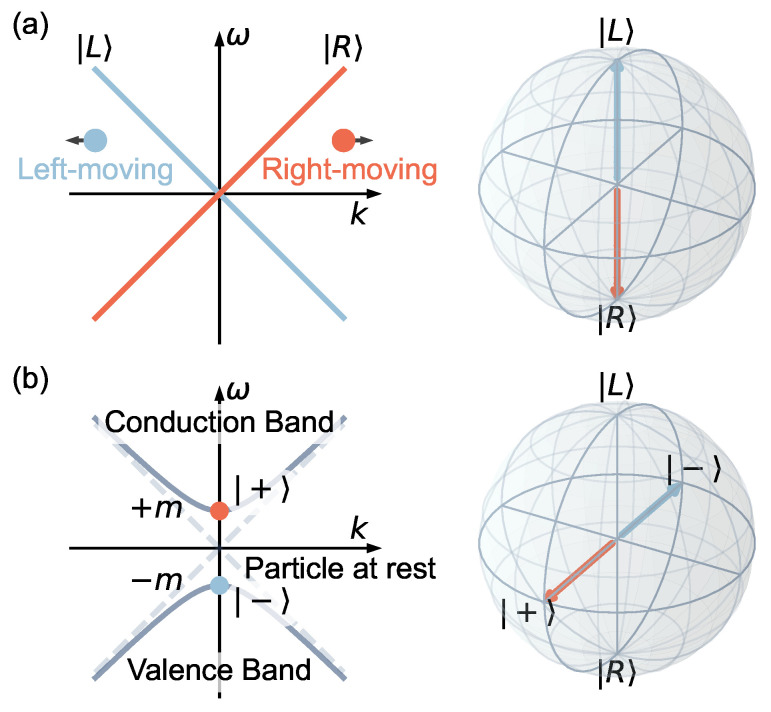
**Dispersion relation, avoided crossings, and qubits.** (**a**) The linear dispersion relation of the left-moving and right-moving massless Fermions. These two states are labeled by |L〉 and |R〉, respectively. They can be mapped onto a qubit, represented by the north and south poles on a Bloch sphere. (**b**) The dispersion relation of massive Dirac Fermion. Once the coupling is introduced, the dispersion relation exhibits an avoided crossing. The two states represent the particle at rest with ±m eigenenergy are represented on the Bloch sphere along the ±X axis.

**Figure 2 entropy-25-01523-f002:**
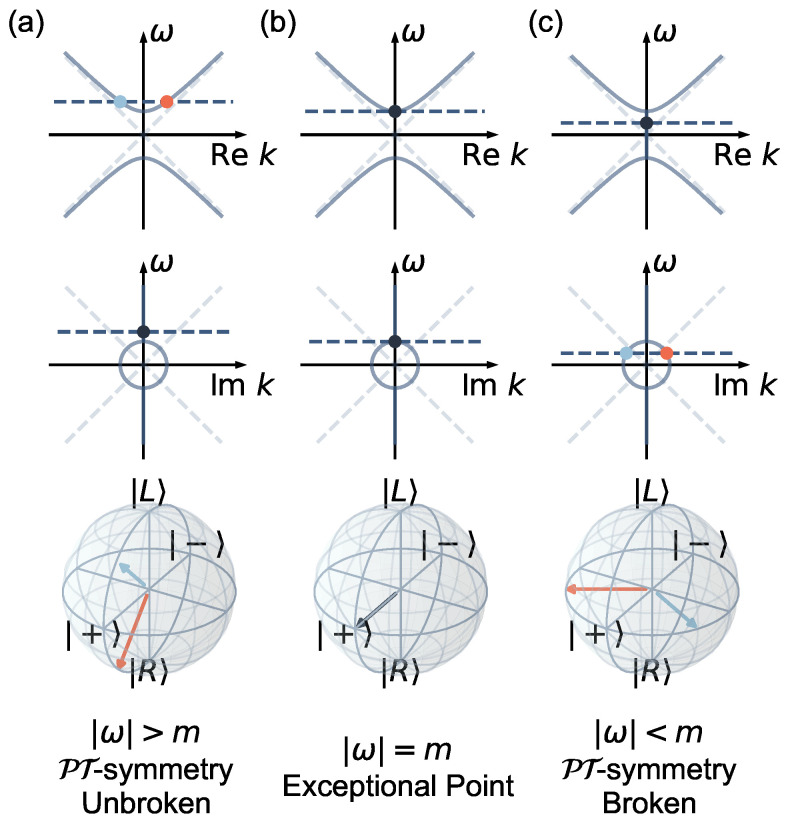
**PT-symmetry.** The parameter space is categorized by the status of the PT-symmetry. In each case, the horizontal line represents the eigenenergy, and the arrows represent the eigenvectors of $H_{\mathrm{eff}}$. (**a**) When |ω|>m, the PT-symmetry is unbroken. The eigenmomenta of $H_{\mathrm{eff}}$ are a pair of real numbers with opposite signs. (**b**) When |ω|=m, the system is at the exceptional point with the eigenmomenta coalescing to 0. (**c**) When |ω|<m, the PT-symmetry is broken. The eigenmomenta of $H_{\mathrm{eff}}$ are a pair of purely imaginary numbers with opposite signs.

**Figure 3 entropy-25-01523-f003:**
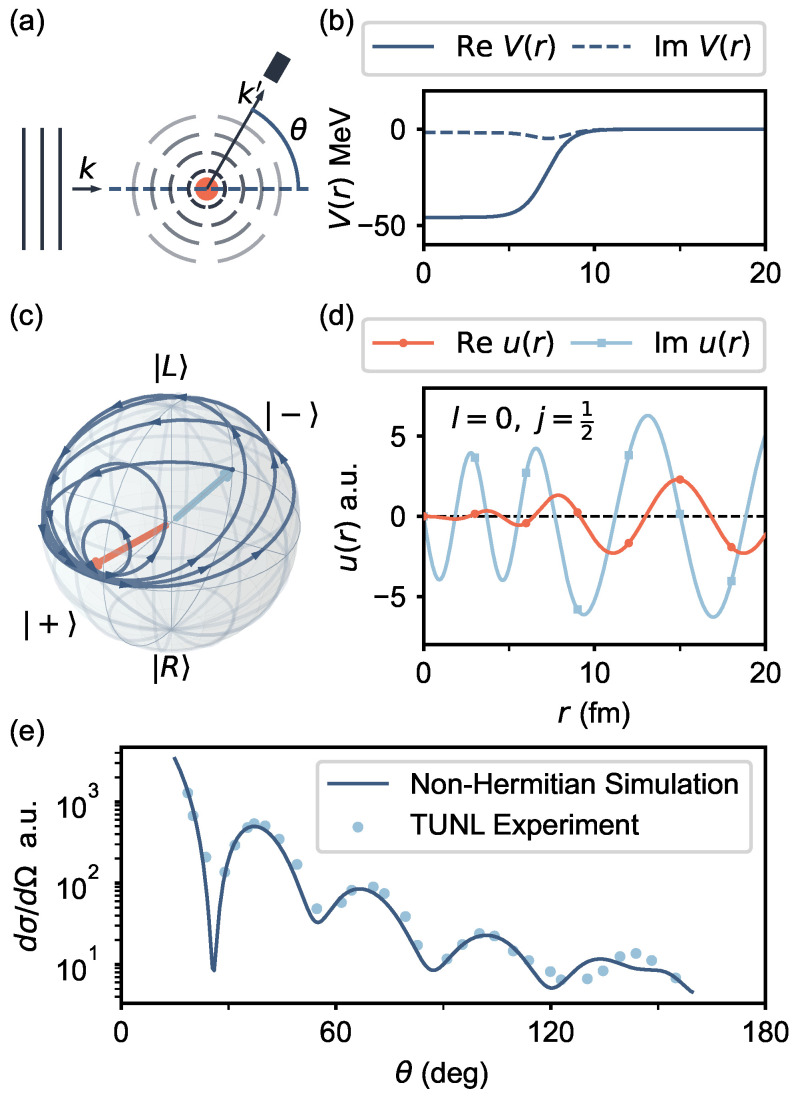
**Simulating scattering phase shifts with qubit dynamics.** (**a**) The scattering of a neutron in plane wave state given by wavevector k scattering off of a nucleus can be expressed in terms of the radial wavefunction for the scattering state $u_{lj}\left(r\right)$. By mapping the radial wavefunction to the components of a spinor describing a qubit, the spatial solution $u_{lj}\left(r\right)$ of the scattering problem corresponds to the time evolution |ψ〉(t). (**b**) The potential function $V_{lj}\left(r\right)$ for ${}^{208}$Pb atomic nucleus given by CH89. (**c**) The trajectory on the Bloch sphere. The vectors have been normalized to make them stay on the Bloch sphere. (**d**) The scattering wavefunction for the lowest partial wave channel ($l=0,j=\frac{1}{2}$). (**e**) Unpolarized differential cross section calculated from the non-Hermitian Hamiltonian compared to the experimental result.

**Figure 4 entropy-25-01523-f004:**
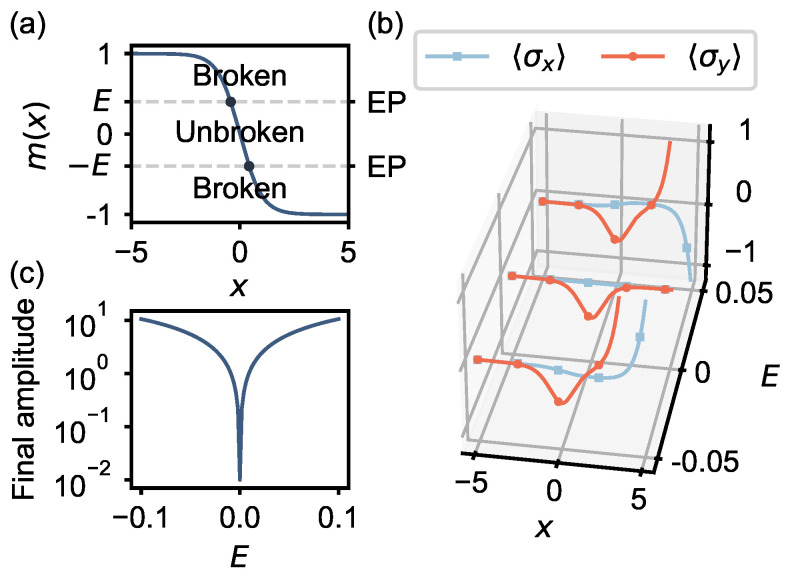
**Majorana zero mode.** (**a**) The spatially dependent mass m(x). When |m|>|E| (|m|<|E|), the system is in the PT-symmetric broken (unbroken) phase. When m=±E, the system is at the EP. (**b**) The wavefunction of the Majorana zero mode given by the $\langle \sigma_{x}\rangle ,\langle \sigma_{y}\rangle$ Bloch components for three different trial energy eigenvalues *E*. The trace for $\langle \sigma_{z}\rangle =0$ is unshown. (**c**) The final amplitude of the quantum state at x=5.

## Data Availability

No new data were created or analyzed in this study. Data sharing is not applicable to this article.

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
