# Peer review of "Parity-Time Symmetric Holographic Principle"

_entropy, 2023, doi:10.3390/e25111523_

Round 1

Reviewer 1 Report

Comments and Suggestions for Authors

The paper considers mapping of the time evolution of a Hermitian particle to the time evolution of a non-Hermitian two by two matrix. The authors then analyze the resulting equation of motion in terms of non-Hermitian quantum mechanics.

I do not recommend its publication in the present form because I have a couple of questions.

First, it seems the first mapping, in which the time derivative and the space derivative are switched, is qualitatively different from the second mapping, in which the second-order differential equation is rewritten as a simultaneous equation of the first-order differential equations. The first one is often done in solving one-dimensional tight-binding model in terms of the transfer matrix. It is called the shooting method. The second one is also often used, e.g. in solving Newton's equation of motion of a pendulum with a resistance proportional to the velocity. The exceptional point indeed corresponds to the critical decay.It seems the only new point is the one on Page 7. The authors might want to explore more deeply into the argument there.

As another comment, I believe the matrix in Eq. (2) is not a usual PT-symmetric matrix. It is simply a T-symmetric matrix. The structure of the eigenvalues of Eq. (3) occurs in any cases with the product of anti-linear symmetry and multiple linear symmetries. In this particular case, I believe it is only the T-symmetry (an anti-linear symmetry) times the identity, not the P-symmetry in the usual sense of flipping up and down.

Author Response

The paper considers mapping of the time evolution of a Hermitian particle to the time evolution of a non-Hermitian two by two matrix. The authors then analyze the resulting equation of motion in terms of non-Hermitian quantum mechanics.

I do not recommend its publication in the present form because I have a couple of questions.

First, it seems the first mapping, in which the time derivative and the space derivative are switched, is qualitatively different from the second mapping, in which the second-order differential equation is rewritten as a simultaneous equation of the first-order differential equations. The first one is often done in solving one-dimensional tight-binding model in terms of the transfer matrix. It is called the shooting method. The second one is also often used, e.g. in solving Newton's equation of motion of a pendulum with a resistance proportional to the velocity. The exceptional point indeed corresponds to the critical decay.It seems the only new point is the one on Page 7. The authors might want to explore more deeply into the argument there.

Response: in our view the impact of our paper is to relate these two qualitatively different mappings to the physics of exceptional points and non Hermitian Hamiltonians. Hence including both examples and the generalization on page 7 form the strength of the manuscript. While it would be possible to expand the discussion of this generalization to show explicit connections to PT symmetry, we found that these arguments were too technical, and ultimately took away from the simplicity of our manuscript. Given this, we are reluctant to add additional explorations here, which we also find challenging to do given the short resubmission window.

As another comment, I believe the matrix in Eq. (2) is not a usual PT-symmetric matrix. It is simply a T-symmetric matrix. The structure of the eigenvalues of Eq. (3) occurs in any cases with the product of anti-linear symmetry and multiple linear symmetries. In this particular case, I believe it is only the T-symmetry (an anti-linear symmetry) times the identity, not the P-symmetry in the usual sense of flipping up and down.

Response: We agree that the matrix in Eq.2 is not of the usual PT-symmetric matrix form, however it does have underlying symmetries which can be shown to be PT symmetric. We felt that the details of these arguments were too lengthy and detailed to include in this short manuscript. Indeed the eigenvalue structure of Eq. 3 can occur in many cases, and really the essential point is that of the “PT symmetry breaking transition”.

We have elaborated on this in the manuscript.

Reviewer 2 Report

Comments and Suggestions for Authors

In this article, the authors present a new way of looking at the traditional method of solving time-independent Schrodinger equations on an unbounded continuum line where the eigenstates correspond to those solutions that, starting from a small value near the left-boundary become square-integrable (k real) instead of diverging (k complex) near the right-boundary. By mapping the single coordinate along which this procedure is carried out to time, the eigenvalue problem can be mapped onto quantum system's time evolution. The authors show this by using two problems. The results in the paper are correct, and the insight - mapping the spatial profile with real or complex k-values to temporal profile with real or complex E-values - allows a clear connection with non-Hermitian, PT-symmetric Hamiltonians. I have only a few comments:

1. The idea of using the time-evolution of a qubit for spatial evolution has been used in the literature, see: PRX 2, 040319 (2021) and a number of related works. The authors should comment on this. 

2. The method of getting +ve or -ve divergent states that bracket a particular Eigen-energy (Figure 4b) has been fairly well-known in the literature. See, for example, Feynman Lectures Volume 3, Chapter 16, figures 16-6/16-8. The authors should mention this to put their theoretical work in context. 

3. The claim for N-dimensional Hermitian to N-1 dimensional non-Hermitian is a bit overstated. Since only one spatial dimension can be transmuted to the time-evolution aspect, it leaves the question of other N-1 dimensional potentials and it is unclear how open system is "an efficient resource for quantum simulation". The authors should clarify this. 

After these changes are taken into account, I recommend publication of the manuscript. 

Author Response

In this article, the authors present a new way of looking at the traditional method of solving time-independent Schrodinger equations on an unbounded continuum line where the eigenstates correspond to those solutions that, starting from a small value near the left-boundary become square-integrable (k real) instead of diverging (k complex) near the right-boundary. By mapping the single coordinate along which this procedure is carried out to time, the eigenvalue problem can be mapped onto quantum system's time evolution. The authors show this by using two problems. The results in the paper are correct, and the insight - mapping the spatial profile with real or complex k-values to temporal profile with real or complex E-values - allows a clear connection with non-Hermitian, PT-symmetric Hamiltonians. I have only a few comments:

  1. The idea of using the time-evolution of a qubit for spatial evolution has been used in the literature, see: PRX 2, 040319 (2021) and a number of related works. The authors should comment on this. 

Response: We appreciate that the reviewer has pointed out this paper and have cited it in our revision

  1. The method of getting +ve or -ve divergent states that bracket a particular Eigen-energy (Figure 4b) has been fairly well-known in the literature. See, for example, Feynman Lectures Volume 3, Chapter 16, figures 16-6/16-8. The authors should mention this to put their theoretical work in context. 

Response: indeed this is the basis of the well known “shooting” method mentioned by reviewer #1. We have included a comment to this effect and include a citation to the Feynman lectures at that point in the text. 

  1. The claim for N-dimensional Hermitian to N-1 dimensional non-Hermitian is a bit overstated. Since only one spatial dimension can be transmuted to the time-evolution aspect, it leaves the question of other N-1 dimensional potentials and it is unclear how open system is "an efficient resource for quantum simulation". The authors should clarify this. 

Response: We are happy to clarify this point. For example, if we are interested in simulating the ground states of a MxM lattice of particles (e.g. qubits) then nominally we would neen M^2 qubits. With this approach, we can reduce the problem to the situation of M qubits and time evolution, which is a huge reduction in resources for quantum simulation. We now specify this more clearly in the manuscript. 

After these changes are taken into account, I recommend publication of the manuscript. 

Response: We appreciate the reviewer’s positive assessment.

Reviewer 3 Report

Comments and Suggestions for Authors

    In this submission, the authors delve into an analogy between unitary spatial motion and non-unitary qubit dynamics. The manuscript harbors intriguing ideas and holds potential for publication in Entropy. However, I have found the manuscript to be somewhat concise, which can make it a bit challenging to read. I would suggest a slight expansion, including a discussion of the efficiency of the proposed method and a more profound physical analysis of the role of PT symmetry in the mapping between spatial and temporal coordinates. Additionally, the conclusions should be extended to better convey the potential implications and significance of this work.

Author Response

In this submission, the authors delve into an analogy between unitary spatial motion and non-unitary qubit dynamics. The manuscript harbors intriguing ideas and holds potential for publication in Entropy. However, I have found the manuscript to be somewhat concise, which can make it a bit challenging to read. I would suggest a slight expansion, including a discussion of the efficiency of the proposed method and a more profound physical analysis of the role of PT symmetry in the mapping between spatial and temporal coordinates. Additionally, the conclusions should be extended to better convey the potential implications and significance of this work.

Response: Indeed we intended for the paper to be concise because we feel that this means that a reader will have time to read the paper, rather than be lost in a verbose discussion. Given the short resubmission deadline presented by the journal, we have expanded the text in a few key areas, largely in response to the other Referee comments.

Round 2

Reviewer 1 Report

Comments and Suggestions for Authors

Perhaps it is better for the authors to repeat the comment added in page 8 every time after relevant examples in the main text.

Author Response

Comments have been added to both of the examples in the main text.